# Management of Neonatal Isolated and Combined Growth Hormone Deficiency: Current Status

**DOI:** 10.3390/ijms241210114

**Published:** 2023-06-14

**Authors:** Stefano Stagi, Maria Tufano, Nicolò Chiti, Matteo Cerutti, Alessandra Li Pomi, Tommaso Aversa, Malgorzata Wasniewska

**Affiliations:** 1Department of Health Sciences, University of Florence, 50139 Florence, Italy; nicolo.chiti@unifi.it (N.C.); matteo.cerutti@unifi.it (M.C.); 2Meyer Children’s Hospital IRCCS, 50139 Florence, Italy; 3Paediatric Unit, Mugello’s Hospital, 50032 Florence, Italy; maria.tufano@uslcentro.toscana.it; 4Department of Human Pathology of Adulthood and Childhood, University of Messina, 98122 Messina, Italytommaso.aversa@unime.it (T.A.); malgorzata.wasniewska@unime.it (M.W.)

**Keywords:** newborn, growth hormone, growth hormone deficiency, congenital hypopituitarism, hypoglycaemia, hormone replacement therapy, cognition disorders/physiopathology

## Abstract

Congenital growth hormone deficiency (GHD) is a rare disease caused by disorders affecting the morphogenesis and function of the pituitary gland. It is sometimes found in isolation but is more frequently associated with multiple pituitary hormone deficiency. In some cases, GHD may have a genetic basis. The many clinical signs and symptoms include hypoglycaemia, neonatal cholestasis and micropenis. Diagnosis should be made by laboratory analyses of the growth hormone and other pituitary hormones, rather than by cranial imaging with magnetic resonance imaging. When diagnosis is confirmed, hormone replacement should be initiated. Early GH replacement therapy leads to more positive outcomes, including reduced hypoglycaemia, growth recovery, metabolic asset, and neurodevelopmental improvements.

## 1. Introduction

Severe congenital growth hormone (GH) deficiency (GHD) in newborns is a rare disease with a reported incidence of between 1:4000–1:10,000 and 1:20,000 newborns [1,2]. Congenital GHD is commonly secondary to anomalies in the morphogenesis and function of the pituitary gland and is often associated with multiple pituitary hormone deficiency (MPHD); nevertheless, GHD can also be isolated (IGHD) [1,2,3,4,5]. IGHD or MPHD may also occur as a result of a genetic syndrome causing abnormalities in extra-pituitary structures that have a common embryological origin to the pituitary gland [4,5,6,7]. Acquired forms of GHD or MPHD may occur because of perinatal or neonatal events [6].

In 5–16% of cases, severe GHD or MPHD has an identifiable genetic basis [2,4]. Generally, genetic mutations in any of the genes involved in the development of the pituitary gland can result in congenital IGHD or MPHD [2,3,4,5,6,7,8,9,10,11,12,13]. However, in most cases the disease’s etiology is unknown, suggesting that other genes may also be implicated [6,8]. It is known that mutations in genes involved in early hypothalamic-pituitary development (such as *HESX1*, *LHX3*, *LHX4*, *SOX2*, *SOX3*, *GLI2*, *OTX2*) are associated with structural abnormalities of the hypothalamic-pituitary axis, midline abnormalities, and extra-pituitary defects, while mutations in genes involved in later stages of development (such as *PROP1* and *POU1F1*) correlate with phenotypes without extra-pituitary defects [14]. We also know that different gene mutations can result in similar phenotypes, and that the same single genetic mutation may be associated with different phenotypes [2,6,8,13]. Recessively inherited or autosomal mutations in the gene coding for growth hormone (*GH1*) or in the gene coding the growth hormone releasing hormone receptor (*GHRHR*) are responsible for IGHD with different phenotypes: type 1A with severe growth failure (*GH1*, autosomal recessive), type 1B with milder growth insufficiency (*GH1*, *GHRHR*, autosomal recessive), type 2 with growth insufficiency, hypoplastic pituitary and potentially other hormone deficiencies (*GH1*, autosomal dominant), and type 3 (other gene mutations X-linked inherited) [2,3,4,5,6,7,8,9,10,11,12,13].

Understanding the different phenotypes, morphological findings and abnormalities is fundamental for establishing the disease’s etiology and for effective follow up [15,16]. The genetic etiologies of GHD/MPHD are summarized in Table 1.
ijms-24-10114-t001_Table 1Table 1Etiology and risk factors of neonatal IGHD/MPHD.CongenitalPerinatal/NeonatalCongenital infectionsBreech delivery/asphyxiaMidline defect syndromes(e.g., septo-optic dysplasia)Neonatal sepsisGene mutations *
* a detailed list of mutations and characteristics of genes involved in pituitary gland development are reported in Table 2. GHD: growth hormone deficiency. MPHD: multiple pituitary hormone deficiency.
ijms-24-10114-t002_Table 2Table 2Mutated genes causing isolated GHD or MPHD, inheritance, radiological features and clinical presentation *.GeneOMIMGenomic LocationInheritanceClinical PresentationRadiological Presentation (MRI)IGHDMPHD**ISOLATED GROWTH HORMONE DEFICIENCY*****GH1**** 13925017q23.31A: AR1B: AR2: ADPostnatal severe (1A) or milder (1B) growth failure and GHDNormal/hypoplastic anterior pituitary (AP) lobe (type 1A and 1B)++ (type II)     *Growth hormone 1*Ectopic posterior pituitary (PP) (type II)***GHRHR**** 1391917p14.3ARGHD symptoms, Milder growth insufficiency (1B)Normal/hypoplastic AP lobe+-     *Growth hormone releasing hormone receptor***NON-SYNDROMIC HYPOPITUITARISM*****PIT1 (POU1F1)**** 1731103p11.2AD, ARHypopituitarism symptomsHypoplastic/normally sized AP lobe ++(GH, PRL, TSH)     *Pituitary specific positive transcription factor 1*No extra pituitary abnormalities***PROP1**** 6015385q35.3ARHypopituitarism symptomsHypoplastic/normal or enlarged AP lobe++(GH, TSH, PRL, LH, FSH, ACTH *)     *Homeobox protein prophet of PIT1*No extra pituitary abnormalities**SYNDROMIC HYPOPITUITARISM*****1. Septo-optic dysplasia (SOD) and its variants******HESX1**** 60180213p14.3AD, ARSOD, IGHD to MPHD with or without optic nerve hypoplasia and or mid-line brain abnormalities, intellectual disabilityNormal/hypoplastic/agenesis AP lobe, ectopic PP lobe, agenesis PS, CC agenesis++(GH, TSH, PRL, LH, FSH, ACTH, DI)     *Homeobox expressed in ES cells 1****SOX2**** 1844293q26.33ADMicro-/anophthalmia, esophageal atresia, genital, dental and brain anomalies, sensorineural hearing loss, micropenis, intellectual disabilityHypoplastic AP lobe, eutopic/ectopic/not visible PP lobe, hypothalamic hamartoma++(LH, FSH, GH)     *Sex determining region Y box 2****SOX3**** 313430Xq27.1X-LinkedCraniofacial abnormalities with or without intellectual disability, hearing impairment.Hypoplastic AP lobe, agenesis/thin PS, CC abnormalities++(GH, TSH, ACTH, LH, FSH)     *Sex determining region Y box 3****OTX2**** 60003714q22.3ADMicro/anophthalmia, seizures, brain malformations, intellectual disability, microcephaly, cleft palateHypoplastic/normal AP lobe, agenesis PS, Chiari I malformation++(GH, TSH, LH, FSH ACTH)     *Orthodenticle homeobox 2****PAX6**** 60710811p13ADMidline craniofacial malformations, ophthalmologic abnormalitiesHypoplastic AP lobe++(GH, ACTH, LH, FSH)     *Paired Box Gene 6****BMP4**** 11226214q22.2ARMacrocephaly, mild psychomotor retardation, skeletal malformations, anophthalmia/microphthalmiaHypoplastic AP lobe, ectopic/not visible PP lobe, CC abnormalities-+     *Bone morphogenetic proteins****FGFR1**** 1363508p11.23ADSOD, midline craniofacial and hand malformations, seizures, Kallmann syndromeNormal or hypoplastic AP lobe, ectopic/eutopic PP lobe, normal/thin/agenesis PS, CC agenesis-+(GH, TSH, ACTH, LH, FSH, DI)     *Fibroblast growth factor receptor 1****ARNT2**** 60603615q25.1AREye malformations, microcephaly, renal abnormalities, seizuresHypoplastic AP lobe, ectopic PP, thin PS, CC abnormalities++(DI, ACTH, GH, TSH)     *Aryl hydrocarbon receptor nuclear translocator 2****PROKR2**** 60712320p12.3AD, ARNeonatal hypoglycemia, micropenis, SOD, Hirschsprung disease, microcephaly, epilepsyHypoplastic AP lobe, ectopic/eutopic PP lobe, agenesis PS, hypoplastic CC + °+(GH, TSH, ACTH)     *Prokineticin receptor 2****2. Holoprosencephaly******GLI2**** 1652302q14.2ADHoloprosencephaly, anophthalmia, cleft lip/palate, midline malformations, imperforate anus, renal agenesisAP hypoplasia, ectopic/not visible PP lobe ++(GH, TSH, ACTH, LH, FSH)     *Zinc finger protein 2***FGF8*** 60048310q24.32AD, ARHoloprosencephaly, SOD, Kallmann Syndrome, Moebius syndrome, microcephaly, spastic diplegiaEnlarged/normal AP lobe, eutopic PP lobe++(LH, FSH, TSH, ACTH, DI, GH ^)      *Fibroblast growth factor 8***3. Pituitary stalk interruption syndrome*****GPR161**** 6122501q24.2ARFacial (congenital ptosis, alopecia) and hands (syndactyly, nail hypoplasia), dysmorphismsHypoplastic AP lobe, ectopic PP lobe, pituitary stalk interruption syndrome described+ °+(GH, TSH, ADH)     *G Protein-Coupled Receptor 161****PROKR2**** 60712320p12.3AD, ARNeonatal hypoglycemia, micropenis, SOD, Hirschsprung disease, microcephaly, epilepsyHypoplastic AP lobe, ectopic/eutopic PP lobe, agenesis PS, hypoplastic corpus callosum + °+(GH, TSH, ACTH)     *Prokineticin receptor 2****OTX2**** 60003714q22.3ADMicro/anophthalmia, seizures, brain malformations, intellectual disability, microcephaly, cleft palateHypoplastic/normal AP lobe, agenesis PS, Chiari I malformation++(GH, TSH, LH, FSH ACTH)     *Orthodenticle homeobox 2***4. Other syndromes*****CHD7**** 6088928q12.2ADCHARGE syndrome (Coloboma of the eye, Heart defects, Atresia of the choanae, Retardation of growth and development, Genital hypoplasia and Ear and hearing abnormalities)AP hypoplasia++(GH, TSH, FSH, LH)     *Chromodomain Helicase DNA Binding Protein 7****GLI3**** 1652407p14.1ADPallister–Hall syndrome: polydactyly, bifid epiglottis, hypothalamic hamartoma, pituitary dysfunction, imperforate anus.Hypothalamic hamartoma, AP hypoplasia++(GH, TSH, LH, FSH, ACTH)     *Zinc finger protein 3****IGSF1**** 300137Xq25X-linkedMacroorchidism, delay in pubertyNormal+ **+(GH, TSH, PRL)     *Immunoglobulin superfamily 1****LHX3**** 6005779q34.3ARSpine abnormalities (short rigid cervical spine), variable degrees of sensorineural hearing lossEnlarged/normal/hypoplastic AP lobe++(GH, TSH, LH, FSH, PRL)     *LIM/homeobox protein 3****LHX4**** 6021461q25.2ADCerebellar abnormalitiesEnlarged//hypoplastic AP lobe, agenesis PS, pituitary cysts, small sella turcica, cerebellar anomalies ++(GH, TSH, ACTH)     *LIM/homeobox protein 4****NFKB2**** 16401210q24.32 ADVariable Immune deficiencyEnlarged/normal/hypoplastic AP lobe, ++(ACTH, GH, TSH)     *Nuclear Factor Kappa-B, Subunit 2****PITX2**** 6015424q25ADAxenfeld—Rieger syndrome: anterior eye chamber, dental hypoplasia, craniofacial dysmorphism, protuberant umbilicusHypoplastic AP lobe, hypoplasia of sella turcica++(GH, LH, FSH)     *Paired-Like Homeodomain Transcription Factor 2****CDON**** 60870711q24.2ADHoloprosencephaly; possibly congenital heart disease, renal dysplasia, radial defects, gallbladder agenesisHypoplastic AP lobe, ectopic/eutopic PP, pituitary stalk interruption syndrome described++(GH, TSH, ACTH)     *Cell adhesion molecule related/down regulated by oncogenes****KCNQ1**** 60411511p15.5ARGingival fibromatosis, mild craniofacial dysmorphic features, short QT syndromeNormal/small hypophysis, thin stalk++(GH, TSH, LH, FSH, ACTH)     *Potassium Voltage-Gated Channel Subfamily Q Member 1****RAX**** 60188118q21.32ARAnophthalmia, microphthalmia and palatal anomalies (bilateral cleft lip and palate)Aplastic pituitary-+(GH, TSH, LH, FSH, ACTH, DI)     *Retina and Anterior Neural Fold Homeobox Gene***ROBO1*** 6024303p12.3ADEye anomalies (strabismus, ptosis)Small/absent AP lobe, ectopic or absent PP lobe, interrupted or absent stalk+ °+(GH, TSH)     *Roundabout Guidance Receptor 1****MAGEL2**** 60528315q11.2ADHypotonia, obesity, developmental delay, contractures and dysmorphismsSmall PP lobe, thin CC, optic nerve hypoplasia++(GHD, ACTH, ADH)     *Mage-Like 2****L1CAM**** 308840Xq28XLRArthrogryposisPartial agenesis of CC+-     *L1 Cell Adhesion Molecule****RNPC3**** 6180161p21.1ARTypical phenotypic features of GHDPituitary hypoplasia+-     *RNA Binding Region (RNP1, RRM) Containing 3****TCF7L1**** 6046522p11.2ADSODAbsent PP lobe, AP hypoplasia, optic nerve hypoplasia, partial agenesis of CC, thin anterior commissure + °+ °     *Transcription Factor 7 Like 1****TGIF1**** 60263018p11.31ADHoloprosencephaly, midline cranial malformations Hypoplastic AP lobe, ectopic PP lobe++     *TGFB Induced Factor Homeobox 1****SIX3***
2p21




***FOXA2**** 60028820p11.21DeletionCongenital hyperinsulinism and hypoglycemiaSmall shallow sella, ectopic PP lobe, interrupted or absent stalk++(GH, TSH, ACTH)     *Forkhead Box A2***TBC1D32*** 6158676q22.31AROro-facial-digital syndrome: retinal dystrophy, developmental delay, facial dysmorphisms Hypoplastic AP lobe, ectopic or absent PP lobe, CC agenesis++     *TBC1 Domain Family, Member 32***EIF2S3*** 300161Xp22.11XLRMEHMO syndrome: profound intellectual disability, microcephaly, growth delay, hypogenitalism, obesity, early-onset diabetes, epilepsyHypoplastic AP lobe, white matter loss++(GH, TSH)     *Eukaryotic Translation Initiation Factor 2, Subunit 3****IFT172**** 6073862p23.3ARRetinopathy, metaphyseal dysplasia, and hypertension with renal failureHypoplastic AP lobe, ectopic PP lobe+-     *Intraflagellar Transport 172****LAMB2**** 1503253p21.31AROptic nerve hypoplasia, focal segmental glomerulosclerosisHypoplastic AP lobe+-     *Laminin, Beta-2*MRI: magnetic resonance imaging; IGHD = isolated growth hormone deficiency; MPHD = multiple pituitary hormone deficiencies; AP: anterior pituitary; PS = pituitary stalk; PP = posterior pituitary; SOD = septo optic dysplasia. XLR: X-linked recessive * https://www.genecards.org/ (accessed on 10 June 2023). * Evolving ACTH deficiency with time; ^ Borderline peak GH concentration; ° Rarely reported in the literature; ** transient/partial.


The aim of this review is to describe the etiology, clinical signs and symptoms, and diagnosis of isolated GHD and MPHD. We highlight the importance of introducing effective therapy during the neonatal period.

## 2. Clinical Presentation

GHD can be isolated (IGHD) or associated with other hormone deficiencies (MPHD). Pituitary deficiencies can involve all the pituitary hormones, including the antidiuretic hormone (ADH) responsible for central or neurogenic diabetes insipidus (DI) [4,7].

Patients have varied and heterogeneous symptoms, according to the severity and number of hormones affected (Table 3). The most common symptoms in newborns are hypoglycaemia, midline abnormalities, micropenis, and prolonged jaundice with cholestasis. Neurological symptoms and psychomotor delay are possible and range from focal deficits to global developmental delay, depending on the underlying genetical anomaly [17].

During the neonatal period, the most frequent presenting feature of congenital GHD is severe hypoglycaemia [18] persisting beyond three days, often in the absence of hyperinsulinism, which may be associated with seizures leading to potential brain damage [1,2,3,4,5,6,7,8,9,10]. This suggests that during the perinatal period, together with cortisol, GH is essential for the regulation of glucose homeostasis [6]. The incidence of hypoglycaemia does not appear to correlate with birth size or severity of GHD, but with the presence of an additional hormone deficiency, particularly with concomitant adrenocorticotropic hormone (ACTH) deficiency [3].

Another typical clinical feature of GHD is the presence at birth, in affected boys, of a micropenis [19,20,21], defined according to a −2.5 standard deviation (SD) cut-off from the mean value. This may result from IGHD or, more frequently, from combined GH and gonadotropin deficiencies [6]. The micropenis may improve when treatment with GH is initiated, suggesting that GH may be critical in penile growth in foetal and early postnatal life [4,6,7].

Some children with GHD or MPHD present with neonatal cholestasis with normal liver parameters and normal gamma-glutamyl transferase levels [22,23,24,25,26,27], as first described in 1956 [28]; for the review: see [29]. The exact cause of this form of hepatitis is not well understood, although GH has been shown to modulate bile acid synthesis and bile acid secretion [22,23,24,25,26]. Central hypothyroidism and hypocortisolism have also been shown to cause conjugated hyperbilirubinemia [29]. There could be several predisposing factors, such as the immaturity of the hepatic excretory function, a susceptibility to viruses or toxins, and a stereotypic response of the immature hepatocyte to injury [22,29]. Prolonged neonatal jaundice may indicate central hypothyroidism [30] and cortisol deficiency can also cause neonatal cholestatic hepatitis [31]. Cholestasis usually resolves spontaneously during the first few months of life. Hormone replacement with GH, L-thyroxine (L-T_4_), and hydrocortisone, in addition to routine intervention for cholestasis, for example with ursodeoxycholic acid, seems to accelerate recovery [22,24,26].

Intrauterine growth is believed to be independent of GH action since most affected newborns present with normal birth length. Some studies have reported early postnatal growth failure, suggesting that GH may be a significant influence on linear growth in this period [3,10,32,33,34,35,36,37].

## 3. Diagnosis

The diagnosis of GHD in neonates is based on clinical signs (hypoglycaemic seizures, midline defects, micropenis) in combination with hormonal and laboratory parameters indicating a deficiency of the pituitary hormones and/or radiological anomalies of the pituitary gland (Table 3) [1,2,3,4,5,6,7,8,9,10,11,12]. A prompt recognition of affected neonates is fundamental, as a delay in replacement therapy can have serious and even lethal consequences [6].

A careful and detailed medical history is mandatory to obtain information about a possible etiology. Information should be gathered on potential predisposing factors, including parental consanguinity, index cases, traumatic/breech birth, neonatal central nervous system infection, and prenatal or birth asphyxia [38]. In a paper evaluating children with hypopituitarism, 7.7% of patients with isolated GHD had a history suggestive of birth asphyxia [39]. Physical examination is also fundamental; for example, height, weight and head circumference should be measured in newborns. Fontanelle size, eyes, microphallus and undescended testicles in males, cleft palate/lip, hepato-splenomegaly, lymphadenopathy, jaundice, and malformations need to be assessed [39,40,41]. Although the diagnosis of MPHD is typically established during the neonatal period, the initial manifestation may occur later in the form of psychomotor delay. In the absence of signs in the neonatal period, early diagnosis may be missed, which can lead to neurocognitive impairment and neurological sequelae [40]. The thyroid hormone (TH) is critical for normal brain development within the first 3 years of life, and a prompt diagnosis of hypothyroidism means that treatment can be commenced immediately to avoid neurocognitive damage. The data suggest that it is just as important to make an early diagnosis of GHD, so that prompt treatment can be introduced to ensure normal brain development [42].

Conventional GH stimulation tests are not recommended because they can be dangerous under 12 months of age. Measuring random basal GH serum concentrations [4,5,6,7] can confirm diagnosis. In the first week of life, infants have relative hypersomatotropism, with random GH levels higher than older children and adults [5,7,43].

GH can be measured in serum or plasma during the first week of life, and thereafter in the stored newborn screening card [5]. Because of GH assay variability, a random GH value of ≤5 μg/L in the first week of life in a neonate with deficiency of other pituitary hormones and hypoglycaemia or pituitary radiological abnormalities is sufficient to distinguish infants with GHD [32,43,44]. Nevertheless, Binder et al. have suggested an ideal GH cut-off of 7 μg/L during the first week after birth [5].

In children, insulin-like growth factor-1 (IGF-1), and insulin-like binding protein-3 (IGFBP-3) are commonly used markers of GH secretion. In the neonatal period, Jensen et al. [45] reported that low serum levels of these markers, below 2SD for days of life, have a high sensitivity (90% for IGF-1 and 81% for IGFBP-3), suggesting that both IGF-1 and IGFBP-3 can be utilized as auxiliary diagnostic tools for GHD [32,43,44].

In the hypothalamus–pituitary–thyroid axis, central congenital hypothyroidism (CH) may be defined as inadequate TH production caused by quantitative or qualitative thyroid-stimulating hormone (TSH) deficiency, leading to TH deficiency in target tissues [46,47]. Biochemically, the patients show FT_4_ concentrations below the reference range associated with normal, low, or slightly elevated TSH levels [47]. The assumption that central CH may be a mild condition has been refuted, and it is critical that neonates are diagnosed shortly after birth. Unfortunately, the data show that most neonates with central CH are diagnosed late, even though many are hospitalized in the first weeks of life for feeding problems, hypoglycemia, or (prolonged) jaundice [46,47]. At present, only a few newborn screening (NBS) programs detect central CH [47]. It is important to remember that, while making a diagnosis of primary hypothyroidism based on elevated TSH is relatively simple, diagnosing central CH is less straightforward, as it calls for the correct interpretation of FT_4_ concentrations. If serum FT_4_ is clearly below the reference range, signs or symptoms of hypothyroidism are present and/or the patient’s medical history is suggestive of hypothalamic or pituitary damage or disease, diagnosis is easier [47], but in some patients, clear signs or symptoms are absent and the medical history is uninformative [47].

The circadian rhythm is established at two [48] to after six months of age [49]; thus, morning cortisol concentrations are not useful in evaluating ACTH deficiency in newborns [6]. Furthermore, low cortisol concentrations, even during a hypoglycaemia episode, have too low a specificity for a diagnosis of adrenal insufficiency [50], and therefore a dynamic assessment (both a low-dose and standard ACTH stimulation testing using tetracosactide hexaacetate) is mandatory [6]. The correct dose of tetracosactide hexaacetate, the optimal timing of blood samples of cortisol measurements, and the cut-off of the peak cortisol concentration, have not been unequivocally established, but stimulated cortisol concentrations ≥18 mg/dL (497 nmol/L) may be considered as indicative of a normal hypothalamo–pituitary–adrenal axis [51].

Gonadotropic hormone deficiencies are confirmed by low levels of plasma gonadotropins, and, in male infants, of testosterone and inhibin B [4,6,7].

Magnetic resonance imaging (MRI) of the pituitary gland can help identify congenital and structural disorders [52,53,54,55]. With sagittal T1-weighted sequences, the posterior pituitary appears as a hyperintense bright spot, while the anterior pituitary is similar in signal intensity to grey matter [53,54]. The gland is proportionally larger in the neonatal period than in childhood. Normal values for the pituitary gland in newborns are elsewhere summarized [7]. A calibre of the stalk of less than 1 mm at any point is generally considered thin [7,53].

Children diagnosed with GHD during the newborn period have a high incidence of neuroanatomical anomalies of the hypothalamic–pituitary region, with a wide spectrum of variations in pituitary anatomy [52,53,54,55]. The most common radiological findings are ectopic posterior pituitary, hypoplastic or aplastic anterior pituitary and an absent or thin pituitary stalk, and an empty sella, even if patients with a normal MRI have also been reported [52,53,54,55].

In infants with isolated GHD, a normal or hypoplastic pituitary gland, empty sella without anatomical abnormalities of the hypothalamus or pituitary stalk are the most frequent imaging features, while a moderate-to-severe hypoplastic pituitary gland (pituitary height ≤ 3 mm) with ectopic posterior pituitary is more frequent in infants with MPHD [55].

In addition, other cerebral abnormalities, including optic nerve hypoplasia, absent septum pellucidum, absent corpus callosum, or Chiari I malformation, have been associated with MPHD [55].

Finally, molecular studies may be fundamental for correctly evaluating patients [6]. Genetic studies should consider patient history and clinical data, as well as laboratory and radiological findings [7,16].

## 4. Treatment and Follow-Up

A multidisciplinary team, including paediatricians, paediatric endocrinologists, geneticists, radiologists, ophthalmologists, and neurologists, should follow and treat infants with GHD [6]. Careful follow-up is mandatory, considering that additional hormone deficiencies may develop, and suitable hormonal treatments could be necessary [4,6,7].

When MPHD is diagnosed, it is important to measure cortisol levels so that newborns with cortisol deficiency can start oral hydrocortisone [4,6,7]. In the case of stress or significant illness, the dose should be doubled or tripled [4].

Oral L-T_4_ is the specific treatment for central hypothyroidism, starting with 50 μg/m^2^ per day or 6–8 to 10/15 microgram/kg/day with the aim of keeping free thyroxine levels in the upper normal range [4,6,56,57]. After starting treatment, the dose needs to be monitored by measuring free thyroid hormone concentrations. TSH levels should not be monitored as the patients are TSH-deficient [4,6,56]. Higher L-T_4_ doses will be needed in newborns with cholestasis due to malabsorption [57,58]. Attention must be given to patients taking iron, soy, calcium, and anticonvulsants [57,58], that can affect L-T_4_ absorption and thus should not be co-administered [58]. Before starting treatment with L-T_4_, it is extremely important to exclude cortisol deficiency, because L-T_4_ increases the basal metabolic rate, enhancing cortisol clearance with the subsequent risk of precipitating an adrenal crisis [4,6,7].

In cholestatic infants, treatment with L-T_4_ and hydrocortisone could require higher doses due to absorption deficiency; the dose should be reduced when cholestasis improves [59].

The therapeutic approach for gonadotropic hormone deficiencies involves replacing the corresponding sex-steroid, rather than the gonadotropins (testosterone injection, dihydrotestosterone gel application or recombinant human gonadotropin subcutaneous infusion) [4,6,7,60,61].

In newborns diagnosed with GHD, whether IGHD or MPHD, recombinant human GH (r-hGH) replacement should be started. Since diagnosis is often delayed and treatment started after the neonatal period [62], the data in the literature are very limited. Here, we focus on the outcomes of early GH replacement treatment and attempt to determine the best timing and dosage of replacement therapy.

When GHD is the recognized cause of persistent hypoglycaemia, replacement therapy with r-hGH, contributes to hypoglycaemic recovery [63,64,65]. Costa et al. describe a child with CHARGE syndrome in whom GHD was diagnosed in the second month of life due to hypoglycaemic episodes: r-hGH was initiated at day 86 (30 μg/kg/day) suggesting that treatment with GH may restore normal glucose homeostasis rather than maintain normal linear growth [66]. Even in other syndromic conditions, the euglycaemic state can be restored by r-hGH replacement. Bonfig et al. [67] describe a 1.5-month-old girl with Turner syndrome and recurrent hypoglycaemia related to GHD: r-hGH therapy was started at a dose of 25–30 μg/kg/day and subsequently doubled (50 μg/kg/day), until blood glucose was normalized.

Early diagnosis and the fast replacement of r-hGH, in addition to increased energy intake and other counterregulatory hormones, seems to prevent recurrent and prolonged hypoglycaemia, although hypoglycaemia may occasionally present in older children during stressful periods associated with reduced oral intake [4].

When GHD is recognized during the first years of life and r-hGH substitutive treatment is started early, short-term and long-term studies demonstrate a marked catch-up growth and significant height gain [34,42,68,69].

GH replacement does not only have a promoting effect on children’s growth but also important metabolic effects [41]. Even at a young age, GHD may show subtle metabolic changes that can adversely affect their future metabolic and atherogenic profile [42]. On the contrary, early treatment could have metabolic effects similar to those reported in patients with Prader–Willi syndrome; in these patients, early r-hGH treatment, before the age of 2 years, is associated with improvements in body composition, motor function, height, and lipid profiles, compared to those who are untreated [70,71]. Further studies are required to clarify the effect of replacement therapy on metabolic asset in GHD infants.

Considering that GH plays an important role in early brain development, maturation, and function, it can be hypothesized that delaying GH treatment could alter brain growth and cognitive abilities [42,72,73,74]. Children diagnosed with congenital pituitary hormone deficiencies may have lower-than-average cognitive functions, and specific difficulties with perceptual organisation compared to siblings [67]. This could be due to various factors such as hypoglycaemia in early life, thyroxine deficiency, or abnormal central nervous system development [72].

Previous studies have reported that treatment with r-hGH, given at dosage of 0.3 μg/kg/day, in association with psychomotor and cognitive stimulation, clearly improves neurodevelopment in children with cerebral palsy [75,76].

The “plastic” function of GH on neuronal structuring and development should be carefully evaluated when considering whether to start r-hGH treatment early, especially when there is a certain diagnosis of GHD but there are no symptoms of classical hormonal deficiency [42]. Although data on GH dosage are scant, a dosage of 25–50 μg/kg/day has been suggested during the first year of life [36,68,69,70].

Considering the lack of data for the neonatal period, careful monitoring is recommended. A low frequency of side effects in older children (intracranial hypertension, slipped capital femoral epiphysis, scoliosis progression) has been reported when r-hGH is used at a conventional dosage [36].

When ACTH deficiency is suspected, hydrocortisone treatment should be started immediately [4,6,7]. Hydrocortisone is the treatment of choice due to its less potent side effects in terms of growth and bone health compared to other glucocorticoids [4,6,7]. The starting dose is 9–12 mg/m2/day, divided into 3–4 doses; a dose higher than for older infants because neonates have greater cortisol secretion rates [4,6,7]. Prior to hospital discharge, families must be instructed about emergency dosing and dosing during illness or periods of stress when a doubling or even tripling of the normal dose is required [4,6,7]. In cases of emergencies, poor tolerance of oral hydrocortisone, or a suspected adrenal crisis, intramuscular hydrocortisone must be administered (<1 year 25 mg, 1–5 years 25–50 mg, >5 years 100 mg) and oral glucose should also be given to correct any associated hypoglycaemia [4,6,7]. Patients who cannot tolerate oral hydrocortisone require hospital admission for intravenous hydrocortisone (1–2 mg/kg every 4–6 h) [4,6,7]. In patients able to tolerate oral hydrocortisone, a triple or double maintenance dose is recommended that can be tapered to a lower dose after clinical improvement [4,6,7].

It is also important to highlight that cortisol deficiency can mask DI, as cortisol is needed for water excretion. DI may develop after starting treatment with hydrocortisone, and therefore close monitoring of fluid balance and electrolytes is important after starting glucocorticoid therapy [51].

Novel treatments, such as continuous subcutaneous hydrocortisone infusion therapy, which may be difficult in neonates due to limited subcutaneous fat for insertion of the cannula, and sustained release hydrocortisone preparations aimed at mimicking physiological cortisol secretion, may become therapeutic options in the future [77].

In newborn male infants, the aim of androgen treatment is to ensure normal testicular descent, improve penile length, and maximize fertility in later life. This treatment will have to be resumed at the time of puberty. In newborns, early treatment is recommended, ideally between 1 and 6 months of age. Testosterone can be given via intramuscular injections or topical gel [60,78,79,80]. Testosterone injections (cypionate or enanthate) are commenced at a recommended dose of 25 mg every 4 weeks for 3 months. This is followed by clinical evaluation of the stretched penile length. Topical gel containing 5-α Dihydrotestosterone (DHT) is also useful, and the recommended starting dose is 1 application (10 mg) every day for 3 months [78]. The carer who is applying the testosterone gel should wash their hands immediately after administration with soap and water and, if the carer is a female, the use of gloves is recommended. Cryptorchidism increases the risk of testicular neoplasia and reduces fertility potential, therefore surgical correction (orchidopexy) is recommended during the first 2 years of life, ideally by 18 months of age [81]. Treatment with LH and FSH during the neonatal period is under investigation [82,83,84].

## 5. Conclusions

Congenital GHD comprises a spectrum of diseases that may cause an isolated deficiency of GH or may be part of a syndrome of MPHD. Clinical manifestations are variable, and include hypoglycemia, micropenis, and cholestasis, in addition to growth problems that often appear later. Therapeutic management should evaluate the clinical signs and the associated hormone deficiencies.

The aim of our work is to summarize the etiology, clinical presentation, diagnosis and current state of therapy for GHD during the neonatal period. Given the wide spectrum of phenotypes and considering that many of the presenting symptoms are non-specific, identifying infants with congenital hypopituitarism is not always simple.

Nevertheless, early identification of GHD is important, because undiagnosed pituitary hormone deficits can lead to significant morbidity and possible mortality.

Most studies have demonstrated the positive role of r-hGH replacement therapy in children affected by GHD, but data on outcomes, timing, and dosage are scant. We strongly recommend the early introduction of GH treatment in GHD, before growth retardation becomes evident. There are clear metabolic and auxological benefits of early intervention. Early diagnosis and the fast replacement of r-hGH seem to prevent recurrent and prolonged hypoglycaemia. In combination with cortisol, this treatment promotes significant catch-up growth.

We also underline that GH could have a plastic role on neuronal structuring during the first years of life, and that untreated GHD could be damaging to brain structure and psychological and neurological development. Further research into therapies for GHD in the neonatal period should be encouraged.

## Figures and Tables

**Table 3 ijms-24-10114-t003:** Symptoms and signs suggestive for IGHD/MPHD.

Symptoms and Signs	IGHD	MPHD (Hormone Deficit)
Hypoglycaemia (with and without seizures)	√	√ (GH, ACTH, TSH *)
Poor feeding	√	√ (GH, ACTH, TSH)
Poor weight gain	√	√ (GH, ACTH, DI)
Lethargy	√	√ (GH, ACTH, TSH)
Cholestasis	√	√ (GH, ACTH, TSH)
Prolonged jaundice		
Conjugated	√	√ (GH, ACTH, TSH)
Unconjugated	-	√ (TSH)
Hepatitis	√	√ (GH, ACTH, TSH)
Seizures without hypoglycaemia	-	√ (ACTH)
Jitteriness	√	√ (GH, ACTH)
Cryptorchidism/scrotal hypoplasia	-	√ (GH, gonadotropin)
Micropenis	√	√ (GH, gonadotropin)
Breech presentation	√	√ (GH, TSH)
Temperature dysregulation	-	√ (TSH)
Electrolyte abnormalities	-	√ (ACTH)
Haemodynamic instability	-	√ (ACTH)
Respiratory distress	-	√ (ACTH, TSH)
Apnoea	-	√ (ACTH)
Polyuria	-	√ (DI)
Polydipsia	-	√ (DI)
Cyanosis	√	√ (GH, ACTH, TSH ^)
Hypotonia	√	√ (GH, ACTH, TSH)
Umbilical hernia	-	√ (TSH)
Bradycardia	-	√ (TSH)
Macroglossia	-	√ (TSH)
Dry skin	-	√ (TSH)
Constipation	-	√ (TSH)
Neonatal and recurrent sepsis	-	√ (ACTH)

* Rarely reported; ^ peripheral cyanosis. GHD: growth hormone deficiency. MPHD: multiple pituitary hormone deficiency.

## Data Availability

Not applicable.

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
