# Peer review of "Management of Neonatal Isolated and Combined Growth Hormone Deficiency: Current Status"

_ijms, 2023, doi:10.3390/ijms241210114_

Round 1

Reviewer 1 Report

Thankyou for a well summarized review on management of GHD and MPHD

The manuscript overall reads well. few revisions that may be considered if seem ok by authors

1. Table 1 and 3 seem redundant- may be removed. The signs and symptoms of GHD and MPHD are very logical standard textbook information

2. The discussion talks about treatment but does not complete the details. Like Till when does testosterone need to continue- details that it may be needed again during puberty. Perhaps authors may summarize this in a table according to each pituitray hormone deficiency with remarks relevant from the discussion. 

Author Response

Florence, 11 June 2023

 Manuscript ID ijms-2414704 entitled “Management of neonatal isolated and combined growth hormone” by Stefano Stagi, Maria Tufano, Nicolò Chiti, Matteo Cerutti, Alessandra Li Pomi, Tommaso Aversa, Malgorzata Wasniewska

Dear Reviewer,

thank you again for this opportunity of cooperation.

We have modified the text of the manuscript, following the reviewers’ suggestions and queries.

This is a point-by-point list of changes performed.

Reviewer: Thank you for a well summarized review on management of GHD and MPHD The manuscript overall reads well. few revisions that may be considered if seem ok by authors

Table 1 and 3 seem redundant- may be removed. The signs and symptoms of GHD and MPHD are very logical standard textbook information

Dear reviewer, the authors believe that, given the particularity of the clinical picture, these tables can be of help and would like to leave them.

The discussion talks about treatment but does not complete the details. Like Till when does testosterone need to continue- details that it may be needed again during puberty. Perhaps authors may summarize this in a table according to each pituitray hormone deficiency with remarks relevant from the discussion.

Thank you for the observation. The review is related to the neonatal period. We have rephrasing the text according to your suggestions.

Reviewer 2 Report

My comments are attached above

My comments are attached above

Author Response

Florence, 11 June 2023

 Manuscript ID ijms-2414704 entitled “Management of neonatal isolated and combined growth hormone” by Stefano Stagi, Maria Tufano, Nicolò Chiti, Matteo Cerutti, Alessandra Li Pomi, Tommaso Aversa, Malgorzata Wasniewska

Dear Reviewer,

thank you again for this opportunity of cooperation.

We have modified the text of the manuscript, following the reviewers’ suggestions and queries.

This is a point-by-point list of changes performed.

In their review the author correlate the genotypes with the phenotype and hormone abnormalities. This review is intended for neurologists, pediatricians and endocrinologists. The text is well organized and the reference is up to date.

Specific comments:

The manuscript can be improved by English editing.

Thank you for the observation. The text are edited by a native English speaker and we hope that now the paper are more clear.

Table 2: syndromatic not syndromic

Thank you for the observation. The text has been modified accordingly. Sorry for the mistake.

The algorithm in the conclusion should be deleted. The differential diagnosis is clear from the text.

Thank you for the observation. We have deleted the figure as for your suggestion.

The conclusion should be shortened.

Thank you for the observation. We have ameliorated the text and we hope that now is more clear.

The tables should be printed after the references, not in the text.

Thank you for the observation. We have moved the tables after the references.

Reviewer 3 Report

A very interesting presentation of the topic of hypopituitarism in terms of GH secretion in the neonatal period. Draws attention and needs updating of the literature. Items 4.10, 17, 20, 23, 25, 37, 52, 62 in my opinion should be replaced with newer publications. Particularly position 17 from 1988 and position 52 from 1973.

On page 4, references are made to a publication by Bonfig et al on hypoglycemia in a girl with Turner syndrome. This part is not related to the topic of the thesis and should be omitted. Similarly, it should be considered whether citing publications about children with Willi-Prader syndrome is related to the main topic of the publication.

The work contains too high a percentage of autocitation.

Author Response

Florence, 11 June 2023

 Manuscript ID ijms-2414704 entitled “Management of neonatal isolated and combined growth hormone” by Stefano Stagi, Maria Tufano, Nicolò Chiti, Matteo Cerutti, Alessandra Li Pomi, Tommaso Aversa, Malgorzata Wasniewska

Dear Reviewer,

thank you again for this opportunity of cooperation.

We have modified the text of the manuscript, following the reviewers’ suggestions and queries.

This is a point-by-point list of changes performed.

A very interesting presentation of the topic of hypopituitarism in terms of GH secretion in the neonatal period.

Draws attention and needs updating of the literature. Items 4.10, 17, 20, 23, 25, 37, 52, 62 in my opinion should be replaced with newer publications. Particularly position 17 from 1988 and position 52 from 1973.

Thank you for the observation. We modified the work changing the references as possibile.

On page 4, references are made to a publication by Bonfig et al on hypoglycemia in a girl with Turner syndrome. This part is not related to the topic of the thesis and should be omitted. Similarly, it should be considered whether citing publications about children with Willi-Prader syndrome is related to the main topic of the publication.

Dear Reviewer, thank you for the observation. We have rephrasing the text and we hope that the paper are now more clear.

The work contains too high a percentage of autocitation.

Thank you for the observation. We modified the work according your suggestions, deletionf some citations